# Graphene-Based Absorption–Transmission Multi-Functional Tunable THz Metamaterials

**DOI:** 10.3390/mi13081239

**Published:** 2022-08-01

**Authors:** Shulei Zhuang, Xinyu Li, Tong Yang, Lu Sun, Olga Kosareva, Cheng Gong, Weiwei Liu

**Affiliations:** 1Institute of Modern Optics, Nankai University, Tianjin Key Laboratory of Micro-Scale Optical Information Science and Technology, Tianjin Key Laboratory of Optoelectronic Sensor and Sensing Network Technology, Tianjin 300350, China; 2120190312@mail.nankai.edu.cn (S.Z.); m18247639565@163.com (X.L.); yangtong0607@163.com (T.Y.); lusun@nankai.edu.cn (L.S.); liuweiwei@nankai.edu.cn (W.L.); 2Faculty of Physics, Lomonosov Moscow State University, Leninskie Gory, 119991 Moscow, Russia; kosareva@physics.msu.ru

**Keywords:** graphene, terahertz, tunable, multi-functional, metamaterials

## Abstract

The paper reports an absorption–transmission multifunctional tunable metamaterial based on graphene. Its pattern graphene layer can achieve broadband absorption, while the frequency selective layer can achieve the transmission of specific band. Furthermore, the absorption and transmission can be controlled by applying voltage to regulate the chemical potential of graphene. The analysis results show that the absorption of the metamaterial is adjustable from 22% to 99% in the 0.72 THz~1.26 THz band and the transmittance is adjustable from 80% to 95% in 2.35 THz. The metamaterial uses UV glue as the dielectric layer and PET (polyethylene terephthalate) as the flexible substrate, which has good flexibility. Moreover, the metamaterial is insensitive to incident angle and polarization angle, which is beneficial to achieve excellent conformal properties.

## 1. Introduction

In recent years, electromagnetic metamaterials have attracted extensive attention because of their many characteristics that natural materials do not have. They can be used to design high-performance devices such as meta-lenses [1,2], filters [3,4] and absorbers [5,6]. In 2008, Landy et al. [5] proposed and experimentally verified the first metamaterial absorber, which achieved perfect absorption at 11.5 GHz, demonstrating important research significance and practical value. Since then, researchers showed great interest in this ultra-thin perfect absorber. After more than ten years of development, the working frequency band of metamaterial absorber has expanded from microwave band [7,8] to terahertz band [9,10], infrared band [11,12] and even optical band [13,14]. The absorption performance has also developed from single frequency [5] and multi-frequency [15] to broadband absorption [16]. However, most of the reported absorbers are “resonant layer–dielectric layer–ground layer” sandwich structure, that is, the electromagnetic wave within the working band can be absorbed by the absorber, the electromagnetic wave outside the working band will be reflected or scattered by the ground layer, cannot penetrate. This structure limits the application on the occasion where electromagnetic waves are not only absorbed but also transmitted in specific frequency bands. In order to overcome this limitation, people began to study metamaterial absorbers with transmission windows, and some progress has been made. In 2012, Costa et al. designed a transmission-type metamaterial absorber in the microwave band based on the resistive film, realizing the function of low-frequency transmission and high-frequency absorption [17]. In 2014, Lu et al. proposed a metamaterial absorber based on magnetic absorbing material. This absorber has the characteristics of low-frequency transmission and high-frequency broadband absorption [18]. In 2019, Li et al. designed a flexible metamaterial absorber, which achieved broadband absorption of 7.7~12.2 GHz and 90% transmittance at 4.35 GHz [19].

Unfortunately, the above-mentioned absorption–transmission metamaterial absorbers are not tunable, and their performances are fixed once they are manufactured. If the electromagnetic performance needs to be changed, it must be redesigned and processed. Moreover, the operating frequency band of these absorbers is mainly in the microwave frequency band. With the development of radar, communication and detection technology towards high frequency and large bandwidth, microwave metamaterial absorber based on traditional materials can no longer meet the future application requirements, and the absorber in the terahertz band has become a research hotspot and application trend. Graphene is a two-dimensional hexagonal honeycomb structure material composed of single-layer carbon atoms [20], which has magical physical and chemical properties. It is a fascinating THz material, which has the advantages of atomic thickness, easy tunability and high kinetic inductance [21]. Its chemical potential can be dynamically adjusted by applying voltage, which makes it play an important role in the research and development of THz tunable devices [22].

Therefore, we propose a multifunctional tunable THz metamaterial based on graphene. The metamaterial can be regulated by applied voltage, and its electromagnetic properties are insensitive to incident angle and polarization, so it has good conformal properties. Moreover, it uses UV glue as the dielectric layer and PET as the flexible substrate, which has good flexibility. The metamaterial has the functions of THz broadband absorption and narrowband transmission. Through full wave simulation and analysis, the absorbance can be adjusted from 22% to 99% in the range of 0.72 THz~1.26 THz, and the adjustment depth can reach 77%. The transmittance at 2.35 THz is adjustable from 80% to 95%.

## 2. Structure Design and Tunable Mechanism

As shown in Figure 1, the proposed graphene tunable metamaterial has the functions of broad band absorption and narrow band transmission. Therefore, both transmission and absorption should be considered in the design to ensure the performance optimization of the metamaterial. The basic idea is to divide the structural design into two parts: (1) broadband absorption design and (2) transmission design. As shown in Figure 1a, the top layer of graphene consists of a central cross structure and a “ring + symmetric diagonals” structure. The center cross structure can form a narrow absorption spectrum, as shown in the red dotted line, while the “ring + symmetric diagonals” structure can form a wide absorption spectrum, as shown in the black dotted line. By integrating the two structures together and optimizing the size parameters, it is possible to achieve both high absorption and wide bandwidth. In addition, four symmetrical diagonal lines around the ring allow the metamaterial periodic elements to be connected to each other to facilitate the uniform application of external voltage. As shown in Figure 1b, the narrow-band transmission structure must be able to transmit efficiently without reducing the absorption. Therefore, it is necessary to not only form a transmission window with a specific frequency, but also form a magnetic resonance with the top-level graphene structure to ensure the maximum absorption. We constructed bandpass-type frequency-selective surface (FSS) units by opening large cross holes in the metal layer to achieve narrow band transmission, and then added a small cross metal shape into the large cross holes to ensure magnetic resonance with the graphene layer, so as to achieve broadband absorption and transmission functions.

Figure 2 shows the structure and size of the graphene-based tunable metamaterial unit cell. It is composed of four layers: the first layer is a graphene pattern layer, the second layer is a dielectric layer made of UV glue, the third layer is a transmission layer based on a frequency selective surface (FSS) and the fourth layer is a PET substrate. The graphene layer uses a central cross structure and a “ring + symmetrical diagonal” composite structure to achieve broadband absorption. The UV glue layer with a dielectric constant of 2 has good flexibility. In the third layer, the large and small cross-shaped FSS is used to construct the transmission channel, and finally the broadband absorption and narrow-band transmission function is realized.

Figure 3a shows the 3D schematic diagram of the proposed graphene tunable metamaterial. It adopts UV glue as the dielectric layer and PET as the flexible substrate, which has good flexibility. The tunable mechanism can be summarized as follows: external voltage–carrier density–chemical potential–conductivity. As shown in Figure 3b, graphene has a zero band gap structure without external electric field, but the band gap can be introduced under the action of external electric field, and its carrier density will change. The change in the graphene carrier density will lead to the change in graphene chemical potential. Then, the conductivity of graphene can be controlled by adjusting the chemical potential of graphene. The KuBo formula [23] defines the conductivity of graphene:(1)$$\begin{array}{c} \sigma \left(\omega ,\mu_{c},\Gamma ,T\right)=\frac{je^{2}\left(\omega -j2\Gamma \right)}{\pi ћ}\times [\frac{1}{{\left(\omega -j\Gamma \right)}^{2}}\int_{0}^{\infty }\left(\frac{\partial f_{d}\left(\varepsilon \right)}{\partial \varepsilon }-\frac{\partial f_{d}\left(-\varepsilon \right)}{\partial \varepsilon }\right)\varepsilon d\varepsilon \\ -\int_{0}^{\infty }\frac{f_{d}\left(-\varepsilon \right)-f_{d}\left(\varepsilon \right)}{{\left(\omega -j2\Gamma \right)}^{2}-4{\left(\varepsilon /ћ\right)}^{2}}d\varepsilon ]. \end{array}$$

Here, −*e* stands for an electron charge, ω is angular frequency, ћ is the reduced Planck’s constant, Γ is the collision frequency and *µ_c_* is the chemical potential. *f_d_* (*ε*) represents the Fermi–Dirac distribution and can be given by:(2)$$f_{d}\left(\varepsilon \right)={\left(e^{\left(\varepsilon -\mu_{c}\right)/k_{B}T}+1\right)}^{-1}$$
where *k_B_* is Boltzmann’s constant and *T* is the temperature. Figure 3c,d, respectively, show the trends of the real and imaginary parts of graphene conductivity changing with its chemical potential and frequency in the terahertz band. It can be seen that as the chemical potential increases, both the real and imaginary parts of graphene conductivity increase. In addition, with the increase in frequency, the real and imaginary parts will change regularly. This phenomenon is especially obvious in the range of 0~2 THz. Theoretically, the chemical potential *µ_c_* of graphene can be determined by the carrier density *n_s_*:(3)$$n_{s}=\frac{2}{\pi {ћ}^{2}\upsilon_{F}^{2}}\int_{0}^{\infty }\varepsilon \left[f_{d}\left(\varepsilon \right)-f_{d}\left(\varepsilon +2\mu_{c}\right)\right]d\varepsilon$$
where *υ_F_* ≈ 9.5 × 10^5^ m/s is the Fermi velocity. The carrier density of graphene can be expressed as:(4)$$n_{s}=\varepsilon_{0}\varepsilon_{d}U/te$$

Here, *ε*_0_ and *ε_d_* represent the dielectric constants of air and bias layer, respectively, *U* is the applied voltage and *t* is the thickness of the bias layer. According to Equations (3) and (4), the relationship between the applied voltage *U* and the chemical potential *µ_c_* can be obtained as follows:(5)$$U\approx \frac{\mu_{c}^{2}et}{\pi \varepsilon_{d}\varepsilon_{0}{ћ}^{2}\upsilon_{F}^{2}}$$

According to the above analysis, we can change the conductivity of graphene by adjusting the applied voltage, and then regulate the electromagnetic properties of graphene metamaterials.

## 3. Simulation, Analysis and Discussion

As a two-dimensional nanomaterial, graphene can be set as a thin film described by surface impedance in simulation [24,25]. In this paper, we use CST Microwave Studio electromagnetic simulation software for modeling and numerical simulation and set the relevant parameters of graphene as: *T* = 300 K, *µ_c_* = 0.8 eV, Γ = 2.5 THz, where *T* is the temperature, *µ_c_* is the chemical potential of graphene and Γ is the collision frequency, which is related to the processing technology of graphene. The graphene-based metamaterials’ reflectance and transmission can be acquired by simulating the complex frequency-dependent S parameters, *S*_11_ and *S*_21_. Then, the calculation formulas of reflectivity *R*, transmittance *T* and absorptivity *A* are as follows:(6)$$\{\begin{array}{c} R={\left|S_{11}\right|}^{2}\\ T={\left|S_{21}\right|}^{2}\\ A=1-R-T=1-{\left|S_{11}\right|}^{2}-{\left|S_{21}\right|}^{2} \end{array}$$

Figure 4a illustrates the absorption and transmission spectra of the metamaterial when the chemical potential of graphene is 0.8 eV. According to the spectral analysis, the metamaterial can achieve broadband absorption in the range of 0.72 THz~1.26 THz, and has 80% efficient transmission at 2.35 THz. The relative absorption bandwidth (*W_RAB_*) of the absorber can be defined as *W_RAB_* = 2(*f_max_ − f_min_*)/(*f_max_* + *f_min_*). Therefore, it can be calculated that the relative absorption bandwidth is 55%.

Figure 4b shows the absorptivity curves at various graphene chemical potential. It can be seen that as the chemical potential decreases, the absorption rate also decreases, and the maximum absorption rate drops to 22% when the chemical potential decreases to 0 eV. The overall regulation depth reaches 77%, with good tunability. Figure 4c shows the transmittance curves at various graphene chemical potential. It can be seen that the transmittance increases with the decrease in chemical potential. When the chemical potential decreases from 0.8 eV to 0 eV, the maximum transmittance increases from 80% to 95%, which also has good tunable performance. Next, the relationship between the applied voltage and the graphene chemical potential can be calculated according to Equation (5). As shown in Figure 4d, as the chemical potential increases, the required applied voltage also increases. Figure 4e shows the schematic diagram of how the external voltage is applied to the graphene metamaterial. The potential difference in the voltage source is applied through the graphene layer and the frequency selective surface. The graphene layer is connected to the positive electrode of the voltage source, and the frequency selective surface is connected to the negative electrode. When the chemical potential is increased to 0.8 eV, 240 kV voltage shall be applied. According to Equation (5), the voltage can be reduced by decreasing the thickness and increasing the dielectric constant of the dielectric layer. In this work, the dielectric layer is UV glue, its thickness is 50 um and the dielectric constant is 2. If the dielectric constant is increased to 8, the thickness is adjusted to 10 nm, then the modulation voltage can be reduced to 12 V.

In most cases, the electromagnetic wave is not normally incident and the polarization angle will change. In order to realize conformal applications, the electromagnetic properties of metamaterials should be insensitive to incident angle and polarization angle. Therefore, we analyze the absorption and transmission properties of the proposed metamaterial at different incidence angles and different polarization angles. Figure 5 shows the electromagnetic properties of metamaterials at different incident angles. The top illustrations are the schematic diagram of TE (left illustration) and TM (right illustration) electromagnetic wave incident modes. Figure 5a shows the simulation results of absorption rate variation with different incident angles in TE mode. It can be seen that as the incidence angle increases, the absorption bandwidth widens and the overall absorption rate remains above 90%. Figure 5c shows the simulation results of absorption rate changing with different incident angles in TM mode. It can be seen that when the incidence angle increases, the absorption bandwidth is basically unchanged and the overall absorption rate remains stable at more than 90%.

Figure 5b illustrates the simulation results of transmittance changing with different incident angles in TE mode. As you can see, the transmittance remains stable as the angle θ increases from 0° to 20°. When the angle θ increases to 40°, the transmission peak splits into three peaks and their intensity decreases. These peaks are located at 2.04 THz, 2.35 THz and 2.71 THz, respectively. However, the transmittances are above 80%. Figure 5d shows the simulation results of transmittance changing with different incident angles in TM mode. It can be seen that although the transmission decreases with the increase in the angle, the maximum transmission is still above 80%. Therefore, although the performance of the metamaterial will change slightly with the change in the incident angle, the overall absorptivity and transmittance remain at a high level, which can be considered insensitive to the incident angle.

Figure 6 illustrates the electromagnetic properties of metamaterials at different polarization angles. Figure 6a,b show the simulation results of absorption rate and transmittance changing with different polarization angles, respectively. It can be seen that when the polarization angle changes between 0~40°, the absorption and transmission properties of the metamaterials are not affected and remain stable, almost unchanged. This is because the structure we designed has perfect symmetry, so it is insensitive to the polarization angle.

Next, in order to explore the broadband absorption mechanism of the designed metamaterial, according to the equivalent medium theory [26], the equivalent impedance of the metamaterial can be calculated by using the simulated S-parameters, as shown in Figure 7. It can be seen that in the frequency band of metamaterial absorption (absorption rate > 90%), the real part of the equivalent relative impedance approaches 1, and the imaginary part approaches 0. This shows that the metamaterial and free-space achieve good impedance matching in the absorption band. It can be calculated that at the position with the strongest absorption (frequency *f* = 0.97 THz), the real part of the equivalent impedance is 1.03 and the imaginary part is −0.07. Therefore, terahertz wave incident on the metamaterial will hardly be reflected and will eventually be absorbed.

To further analyze the absorption and transmission mechanism of the proposed multifunctional graphene tunable metamaterial, we simulated the current distribution and the electric field distribution at three key frequency points, as shown in Figure 8. These three frequency points are 0.72 THz, 1.26 THz and 2.35 THz, representing two absorption frequency points and one transmission frequency point, respectively. We use color to represent electric field strength and current strength, red for strong electric field and current, blue for weak electric field and current. At 0.72 THz, the graphene structure of the metamaterial unit produces strong electric and magnetic resonance. It should be mentioned that the magnetic resonance is produced by the interaction of the graphene layer and the metal layer on the FSS. Under the action of terahertz radiation, both the top graphene structure and the metal layer of the frequency selective surface will generate induced current, which are in opposite directions and parallel to each other. At this time, it can be seen that magnetic dipoles with opposite directions and parallel to each other are generated. When the magnetic resonance of the magnetic dipoles is the strongest, it will consume a lot of terahertz energy and realize strong absorption. Similarly, at 1.26 THz, the edge of the graphene ring generates strong electric resonance, which is due to the resonators formed between the graphene structures of adjacent cells. The strong electric resonance combined with magnetic resonance also produces strong absorption. In addition, it can be seen that the magnetic resonance of graphene-based metamaterial is very weak at 2.35 THz, so the absorption rate is low at this frequency. However, at 2.35 THz, the layer where the frequency selective surface is located produces a strong electric field intensity, which indicates that the frequency is just in the transmission band, forming a transmission window.

## 4. Conclusions

In summary, we proposed an absorption–transmission multifunctional tunable terahertz metamaterial based on graphene. Its patterned graphene layer enables broadband absorption by introducing multiple resonances, while the frequency selective layer enables transmission in specific bands. The metamaterial has the advantage of conformal because it is flexible and insensitive to incident angle and polarization. Furthermore, its absorption and transmission can be controlled by applying voltage to regulate the chemical potential of graphene. The analysis and calculation results show that the absorption of the metamaterial is adjustable from 22% to 99% in the 0.72 THz to 1.26 THz band, and the transmittance is adjustable from 80% to 95% in 2.35 THz band. Therefore, we believe that the excellent properties of this graphene metamaterial will make it have great application prospects in detection, stealth and communication.

## Figures and Tables

**Figure 1 micromachines-13-01239-f001:**
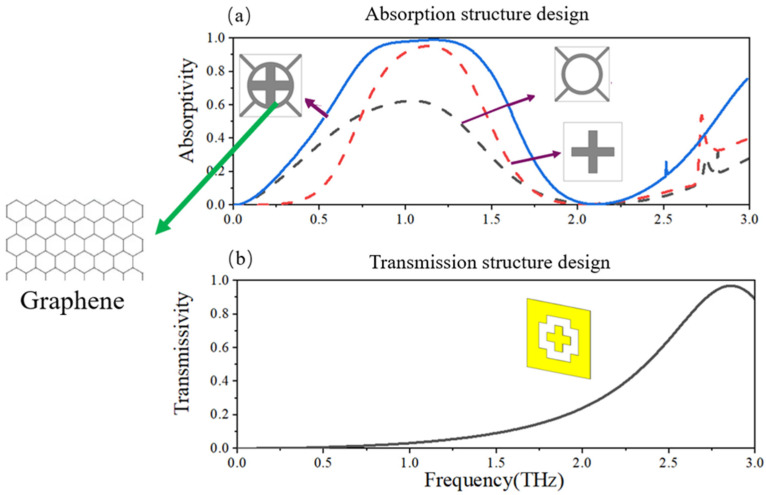
(**a**) The broadband absorption graphene metamaterial structure design; (**b**) The frequency selective transmission structure design.

**Figure 2 micromachines-13-01239-f002:**
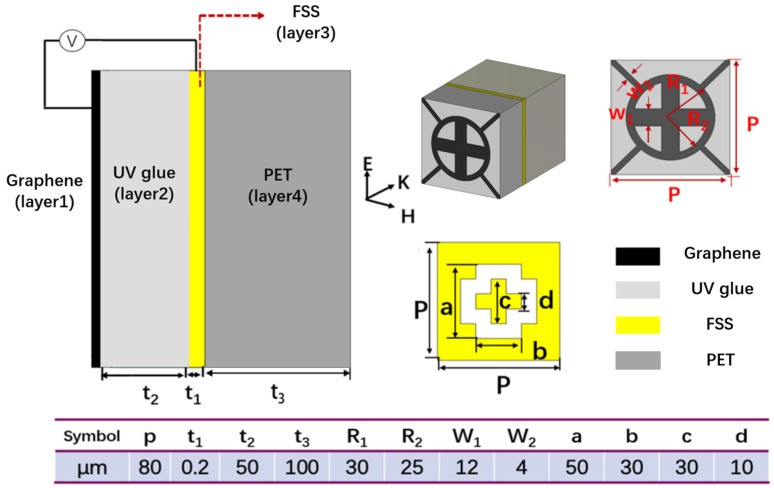
Structure and size of the graphene-based tunable metamaterial unit cell.

**Figure 3 micromachines-13-01239-f003:**
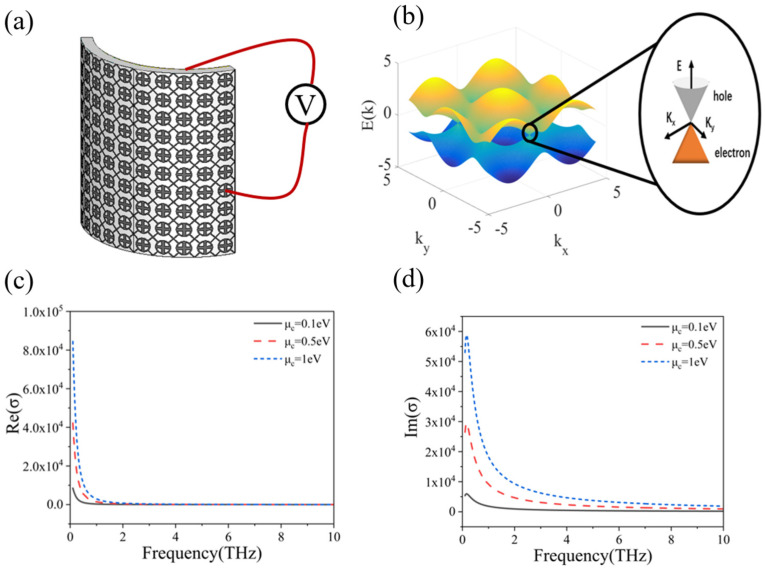
(**a**) Three-dimensional schematic diagram of the flexible graphene tunable metamaterials; (**b**) Schematic diagram of graphene band gap; (**c**) Trends of real part of graphene conductivity with chemical potential and frequency; (**d**) Trends of imaginary part of graphene conductivity with chemical potential and frequency.

**Figure 4 micromachines-13-01239-f004:**
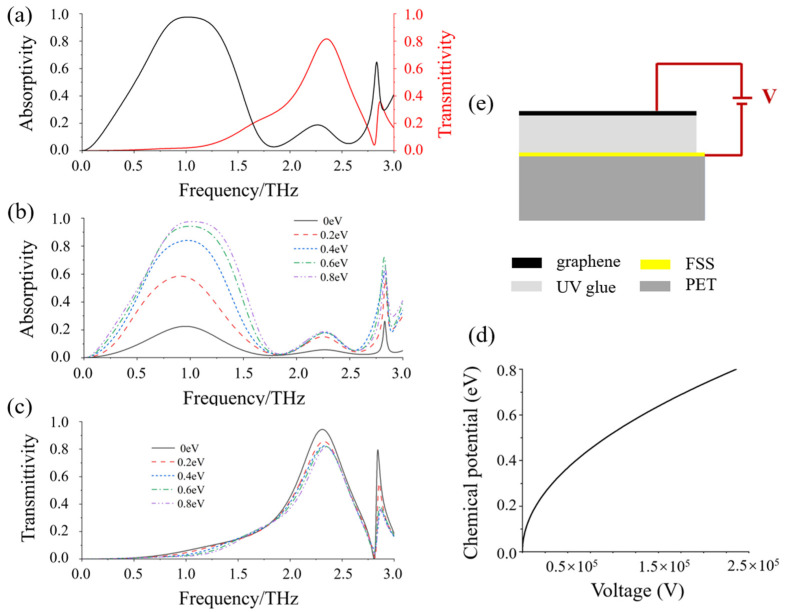
(**a**) The absorption and transmission spectra when the chemical potential is 0.8 ev; (**b**) The absorptivity curves at various graphene chemical potential; (**c**) The transmittance curves at various chemical potential; (**d**) The relationship between the applied voltage and the graphene chemical potential; (**e**) Schematic of how the external voltage is applied to the graphene metamaterial.

**Figure 5 micromachines-13-01239-f005:**
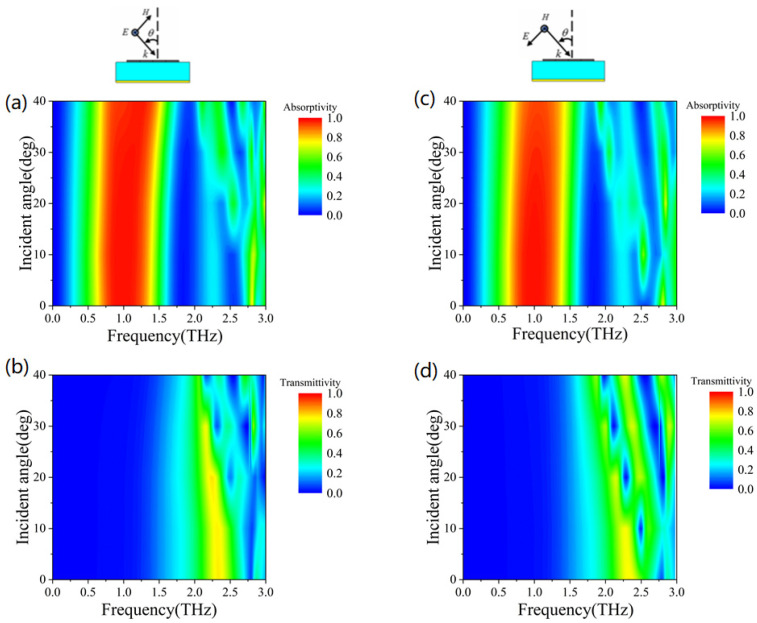
(**a**) The simulation results of absorption rate variation with different incident angles in TE mode; (**b**) Transmittance changes with different incident angles in TE mode; (**c**) Absorption rates changing with different incident angles in TM mode; (**d**) Transmittance changing with different incident angles in TM mode.

**Figure 6 micromachines-13-01239-f006:**
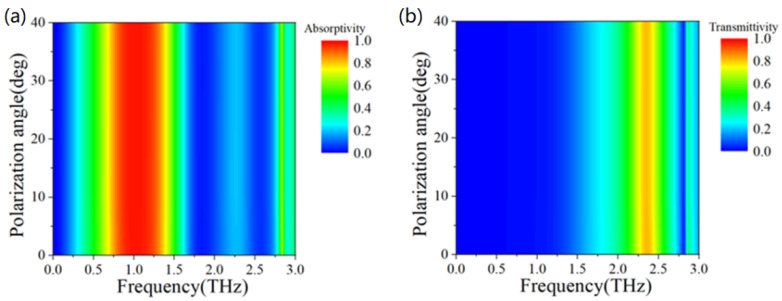
(**a**) The simulation results of absorption rate changing with different polarization angles; (**b**) Transmittance changing with different polarization angles.

**Figure 7 micromachines-13-01239-f007:**
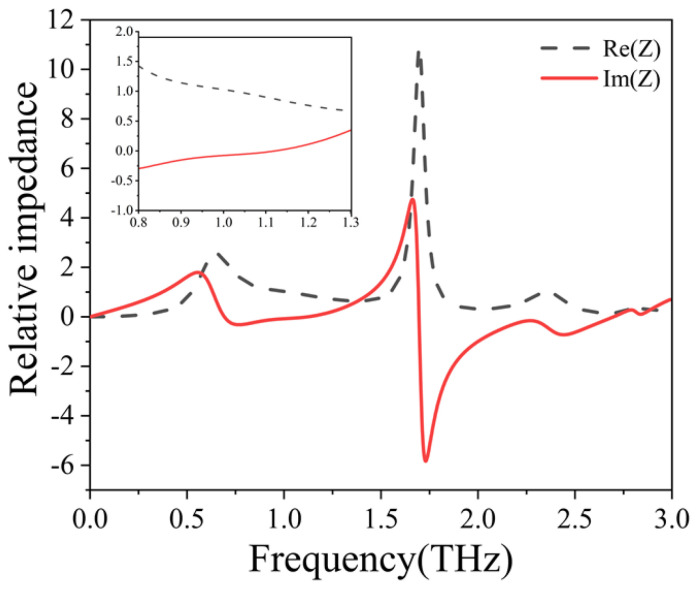
The equivalent relative impedance of the proposed metamaterial.

**Figure 8 micromachines-13-01239-f008:**
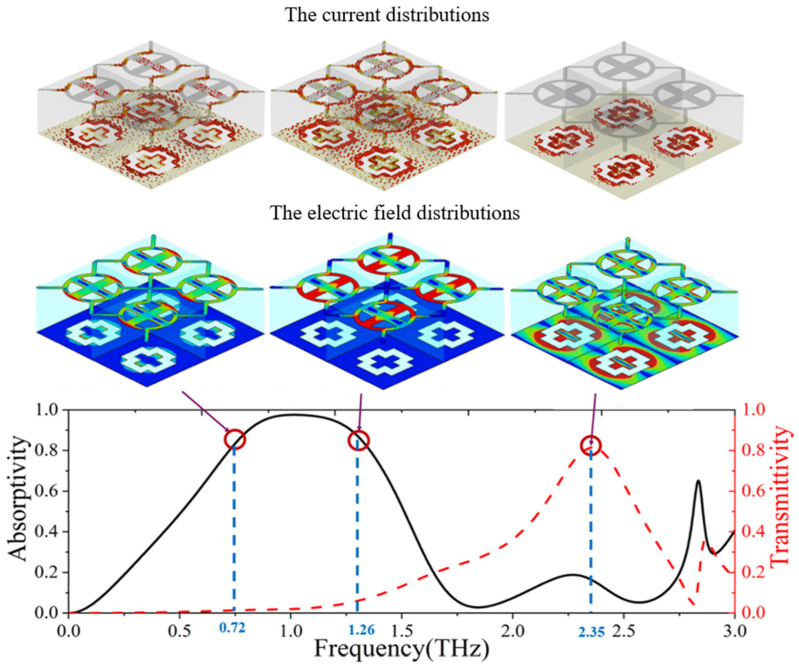
The current distribution and the electric field distribution of the graphene tunable metamaterial at three key frequency points.

## Data Availability

The data that support the plots within this paper are available from the corresponding authors on reasonable request.

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
