# Peer review of "Graphene-Based Absorption–Transmission Multi-Functional Tunable THz Metamaterials"

_micromachines, 2022, doi:10.3390/mi13081239_

Round 1
Reviewer 1 Report
In this paper, the authors report a graphene-based absorption-transmission multifunctional tunable terahertz metamaterial. It is an interesting work. The metamaterial has the conformal properties because it is flexible and insensitive to incident angle and polarization. Furthermore, its absorption and transmission can be controlled by applying voltage to regulate the chemical potential of graphene. The simulations and analysis are carefully performed, and the results look reliable. Therefore, I suggest accepted this manuscript with the following revisions:
1) In this paper, graphene is modulated by voltage, and then THz metamaterials are regulated. What is the modulation speed? There is no relevant introduction in the paper?
2) The paper uses a maximum voltage of 280V for modulation, but high voltages introduce some additional limitations. Suggest to discuss the method of reducing modulation voltage?
3) The paper is too brief about how to apply voltage to metamaterials. Please introduce in detail how to apply voltage, and which layer is the positive and negative electrodes applied to? Why?
4) Are the properties of the proposed graphene-based metamaterials unique in the terahertz band? Can it be applied to other bands? Analysis is recommended.
Reviewer 2 Report
Reviewer Comments
On Manuscript ID: micromachines-1821104 “Graphene based absorption-transmission multi-functional tunable THz metamaterials”
Dear Authors.
Recently, the number of articles has significantly increased, in which various tunable devices of the microwave and terahertz frequency ranges such as filters, EMW absorbers, and so on, using graphene layers as a working medium, are investigated. In most of these works, the same formulas for the complex conductivity of graphene are given and the possibility of controlling the properties of these devices by changing the conductivity is considered. Moreover, as in this manuscript, in most works, the authors limit themselves to modeling in CST Microwave Studio. The results of the practical implementation of devices are, unfortunately, much less common.
I studied the text of the manuscript and the articles those authors cited.
I have several comments on the text of the manuscript under review:
Page 3
Line 89. Dear authors! It is necessary to explain how and due to what interactions magnetic resonance is formed?
Line 96. The layer thicknesses t1 and t2 shown in the figure 2, with t2 greater than t1, do not correspond to the data presented in the table below the figure.
Line 103. Firstly, it is not aluminum, but its oxide, that has such a permittivity.
Secondly, how can the potential difference from the voltage source V through the dielectric be applied to the graphene layer? The graphene layer is one conductive plate of the capacitor, but where is the second conductive plate? How is the conductivity of the graphene layer controlled in such a system?
Line 104. Figure 2 is missing the pet (polyethylene terephthalate) as the flexible substrate layer.
The presence of this layer is stated in the abstract and in the next paragraph.
Page 4.
Line 120. Dear authors! How to understand this proposal: “…Then the chemical potential graphene can be controlled by adjusting the chemical potential of graphene…”?
Page 6.
Line 153. Dear authors! The curve for the transmission coefficient in fig. 4(a) does not correspond to the chemical potential value of 0.8 eV. This can be seen from a comparison with the data in Figure 4(c).
I believe that the manuscript requires careful review of the text, correction of comments, and re-review.
Round 2
Reviewer 2 Report
The authors have corrected all my comments on the text of the manuscript.
However, there is one remark to figure 4 on page 6 of the manuscript.
In Figure 4(c) there is a mistake in the designation of the y-axis. Instead of “transmittance”, the axis is labeled as “absorptivity”.
I believe that the manuscript can be published after correcting this error.
